# Formal Versus Self-Identified Neurodivergence: A Comparative Study in Work Environments

**DOI:** 10.3390/bs15040420

**Published:** 2025-03-25

**Authors:** Jan van Rijswijk, Petru Lucian Curșeu

**Affiliations:** 1Department of Organization, Open Universiteit, 6419 AT Heerlen, The Netherlands; jan.vanrijswijk@ou.nl; 2Department of Psychology, Babeș-Bolyai University, 400015 Cluj-Napoca, Romania

**Keywords:** neurodivergence, self-identification, rationality, decision-making styles, cognitive styles, self-awareness

## Abstract

This study investigated the added value of using self-identification of neurodivergence next to formal diagnosis in predicting cognitive differences. We collected and analyzed data from neurodivergent and neurotypical employees in a sample of 357 participants in 19 organizations across seven industries. Our results are aligned with previous results that support a systematic information processing tendency in highly gifted and autistic employees and decision impulsiveness in those with ADHD and ADD. Supporting previous findings, our results show different cognitive profiles of dyslexia and dyscalculia. Finally, our results show that self-identified neurodivergence adds to the predictive value of formally diagnosed conditions and that in empirical organizational research, self-identified neurodivergence is sufficient to capture the cognitive differentiation tied to neurodivergence.

## 1. Introduction

Although in the past, neurodivergent individuals faced systematic exclusion from the labor force due to their different ways of thinking, nowadays we see a significant shift in recognizing their indispensable role in the workforce, as neurodivergent individuals make up an aggregated 22% of the population ([6]; [30]). Particularly in innovation-oriented organizations, there is a growing acknowledgment of the necessity to accommodate neurodivergent employees and develop inclusive work practices to address the challenges stemming from these conditions in order to take advantage of their unique work perspectives ([16]).

Neurodivergent employees display particular information processing patterns and are thought to have differently wired brains than neurotypical employees ([16]; [25]). Beyond the challenges raised by these distinct cognitive patterns and information processing tendencies, neurodivergent employees also bring unique perspectives that have the potential to enhance creative performance, especially when neurotypical and neurodivergent employees work together and exchange their different perspectives on the task ([29]). However, due to stigma and prejudices associated with neurodivergent conditions, neurodivergent employees often mask their condition and do not seek a formal diagnosis. [8] ([8]) identified three key reasons why many neurodivergent individuals do not have a formal diagnosis: (1) official diagnosis is not accessible due to long waiting lists, a lower ability to navigate the healthcare system, or prohibitive costs, (2) neurodivergent individuals do not seek a formal diagnosis because it brings the risk of social stigmatization, and (3) formal diagnosis is less accurate than self-identification. Due to the personal and social challenges faced throughout life by many neurodivergent employees, their self-awareness concerning their diagnosis is often accurate ([8]; [21]). As such, the prevalence of neurodivergence in organizations could be higher, as employees may choose not to seek a formal diagnosis, yet they are well aware of their neurodivergent condition.

Although the literature claims that neurodivergent employees contribute with their different cognitive styles and competencies to the task ([1]; [4]), we know little about such differences in organizational settings ([28]). Moreover, because scholars who investigate the integration of neurodivergent employees in work environments predominantly rely on formal diagnoses and thereby inadvertently overlook a substantial cohort of undiagnosed neurodivergent individuals who are improperly categorized as neurotypical, it becomes important to understand whether self-identification of neurodivergence adds to the predictive power of formal diagnosis in relation to cognitive differences.

The main aim of our paper is, therefore, to investigate the extent to which self-identified neurodivergence adds to the predictive power of formal diagnosis concerning such differences. More specifically, our study evaluates differences in decision-making styles, rationality, and cognitive styles among neurodivergent and neurotypical employees, and explores the extent to which the self-identified neurodivergent conditions add to the initial predictive power of the formal diagnosis of neurodivergence. By doing so, our paper presents essential insights that can be used to assess neurodivergence in organizational research, empowering scholars to pursue more accurate research on cognitive differences tied to neurodivergence in work settings.

## 2. Theoretical Background

Neurodivergence is marked by distinct cognitive wiring compared to the neurotypical majority ([16]; [25]). This cognitive diversity is a natural outcome of evolutionary progress and results in neurodivergent individuals using information in unique ways that manifest itself in the encoding (i.e., acquiring information), retention (i.e., remembering information), and retrieval (i.e., processing information into new outcomes) stages of information processing ([20]). While detailing each stage for every neurodivergent condition would be exhaustive, briefly describing some variances compared to neurotypical individuals is worthwhile. During the encoding stage, neurodivergent individuals often encounter challenges in gathering information. People with conditions like autism and ADHD may absorb vast amounts of information but can struggle with the interpretation and prioritization of it ([23]). Conversely, dyslexia and dyscalculia primarily hinder the absorption of information, especially in written forms ([17]), and environmental cues can obstruct information retrieval for individuals with a heightened sensitivity that is common for highly gifted and autistic people ([13]; [30]). Information retention is for ADHD-diagnosed people often unstructured as part of the difficulties in making the distinction between relevant and irrelevant information ([25]). However, in many cases of autism, this priority problem leads to high levels of information storage on topics of interest to that person ([23]). The perceived differences in information encoding and retention lead neurodivergent individuals to unexpected outcomes in the information retrieval stage, resulting in both advantages and disadvantages in work settings. For example, the unstructured form of information encoding and retention in the case of ADHD often leads to failure to follow established organizational procedures, yet, at the same time, may also result in creative insights and inspiration for further innovations ([1]). Conversely, autism often leads to information processing difficulties in ever-changing environments, yet these employees often excel in tasks that require great attention to detail ([4]).

Due to the consequences of the unique cognitive wiring of neurodivergent individuals, reflected in different decision-making styles, rationality, and thinking patterns, they often encounter challenges as employees. However, in practice, a significant number of these employees remain undiagnosed ([8]; [18]). Sometimes, individuals display only a spectrum of characteristics of a neurological condition and do not meet the formal criteria for diagnosis. Moreover, research indicates that in certain populations, diagnosticians often lack sufficient knowledge of the characteristics of neurodivergent conditions in these populations, leading to misdiagnosis or a complete lack of diagnosis ([11]). Furthermore, other neurodivergent employees may consciously avoid seeking a diagnosis as a protective measure against potential social stigmatization ([8]; [22]).

The lack of formal diagnoses complicates efforts to conduct comprehensive research on neurodivergence in work environments. We, therefore, set out to explore the extent to which self-identified neurodivergence has additional predictive power to the formally diagnosed neurodivergence on the cognitive functioning of employees, and we take into account differences in decision-making styles, cognitive styles, as well as cognitive competencies. We expect that self-identified neurodivergence adds significantly to the predictive power of the models concerning differences in cognitive functioning between neurotypical and neurodivergent employees.

## 3. Methods

We employed a snowball approach to gather data for our study, reaching out to diverse organizational teams as part of a broader study on cognitive differences in group dynamics. We start by briefing team managers individually on the study’s purpose and procedure, asking for their assistance and consent. Subsequently, with their commitment and approval, we distributed the digital survey by e-mail to the team members. After a few weeks, we sent a short reminder to enhance the volume of our sample. In total, we collected data from 357 participants across 19 organizations in 7 industries (1 organization in banking, 5 in healthcare, 1 in pensions and insurance, 2 in education, 1 in employee recruitment, 7 in IT, and 2 in the construction industry), all located in The Netherlands and Bonaire. The sample included 43 participants who reported being formally diagnosed with a neurodivergence condition and 95 reported that they self-identify with at least one of the neurodivergence conditions. These results are aligned with previous research showing that the incidence of neurodivergence is much higher than what is indicated through formal diagnoses as people avoid the risk of stigmatization tied to a formal diagnosis for a neurodivergent condition or simply because the availability of specialized diagnosing services is low or too expensive ([8]).

Decision-making styles were evaluated using the General Decision-Making Style Inventory ([26]). Five items evaluated each of the following decision-making styles: rational (item example “I double-check my information sources to be sure I have the right facts before making decisions”), intuitive (item example “When I make decisions, I tend to rely on my intuition”), dependent (item example “I often need the assistance of other people when making important decisions”), avoidant (item example “I postpone decision-making whenever possible”), and spontaneous (item example “I often make decisions on the spur of the moment”). Previous research extensively tested the reliability and validity of this questionnaire ([5]; [27]) as an accurate indicator of decision-making tendencies.

Cognitive styles were evaluated using the Verbalizer–Visualizer Questionnaire ([24]), which was extensively used to assess individual preferences in visual versus verbal information processing. For the verbal cognitive style, examples of items include: “I enjoy doing work that requires the use of words”, while for the visual cognitive style, examples of items include “My thinking often consists of mental pictures or images” (answered as true or false).

Rationality was evaluated using the three items of the Cognitive Reflection Test ([9]), which provides an objective measure of a lower-order factor of general mental ability ([3]). Previous results have extensively documented the usefulness of this method for assessing participants’ capacity to engage in analytic and deliberate information processing ([12]; [14]).

Diagnosed neurodivergence was evaluated by asking participants whether they were formally diagnosed with one of the following conditions (“Which of the following conditions have third parties formally diagnosed about you?”, answered as yes or no for the diagnoses autism, ADHD, ADD, DCD, dyslexia, dyscalculia, and high giftedness) and entered in the analyses as a combines dummy variable (1 = at least one presence versus 0 = absence).

Self-awareness of neurodivergence was evaluated by asking participants whether they believe they have one of the neurodivergent traits associated with the following conditions: (“Which of the following conditions do you personally think apply to you?”, answered as yes or no for the diagnoses autism, ADHD, ADD, DCD, dyslexia, dyscalculia, and high giftedness) and entered in the analyses as a combined dummy variable (1 = at least one presence versus 0 = absence).

Education was evaluated as a categorical variable by asking participants to specify their highest completed level of education: (1) vocational education; (2) secondary education; (3) bachelor’s degree from a university of applied sciences; (4) master’s degree from a university of applied sciences or bachelor’s degree from a research university; (5) master’s degree from a research university; (6) and doctoral degree.

To preserve anonymity, we did not collect demographic information, except for education, which was deemed relevant as a control variable because educational achievements can be related to neurodivergent conditions and because education is correlated with cognitive competencies, cognitive functioning, and information processing styles ([10]).

## 4. Results

Means, standard deviations, and correlations among the study variables are presented in Table 1.

We have used stepwise OLS regression analyses to explore the predictive value of the neurodivergent conditions assessed in our research. In the first step of the regression, we entered education as a control variable and the formally diagnosed neurodivergent conditions, except for dyscalculia and DCD, which were only present in the self-identified answer set. Overall, 43 participants (12.1%) reported at least one formally diagnosed neurodivergent condition, while 95 participants (26.7%) reported self-awareness of at least one neurodivergent condition. These percentages are in line with previously reported incidences of neurodivergence in the general population ([30]). Neurodivergent conditions were entered as dummy variables, with the neurotypical employees serving as a general reference category. Because the predictors included in the two steps of the regression analyses were expected to be correlated, we explored the variance inflation scores to check for multicollinearity problems in our analyses. None of the scores exceeded 1.6; therefore, we can conclude that multicollinearity is not a problem in the regression analyses. The results of the regression analyses for the five decision-making styles are presented in Table 2, while the results for rationality and cognitive styles are presented in Table 3.

Education had a negative association with the intuitive style (β = −0.11, *p* = 0.04) and a positive association with the dependent (β = 0.13, *p* = 0.02) and avoidant (β = 0.12, *p* = 0.03) decision-making styles, rationality (β = 0.19, *p* < 0.001), as well as with the verbal cognitive styles (β = 0.14, *p* = 0.01). Self-identified high giftedness had a positive association with the rational decision-making style (β = 0.12, *p* = 0.04) and rationality (β = 0.14, *p* = 0.02). Self-identified autism had a negative association with the intuitive decision-making style (β = −0.22, *p* < 0.001). Formally diagnosed ADHD had a positive association with the spontaneous decision-making style (β = 0.16, *p* = 0.008), a positive association that was observed for self-identified ADHD as well (β = 0.16, *p* = 0.01). Moreover, self-identified ADHD had a positive association with the verbal cognitive style (β = 0.17, *p* = 0.009). Formally diagnosed ADD had a negative association with the rational decision-making style (β = −0.27, *p* < 0.001), an association that was not observed for self-identified ADD (β = −0.23, *p* < 0.001). Self-identified ADD had a positive association with the avoidant (β = 0.14, *p* = 0.01) and with the spontaneous decision-making style (β = 0.12, *p* = 0.03). Formally diagnosed dyslexia had a negative association with intuitive (β = −0.13, *p* = 0.04) and spontaneous decision-making styles (β = −0.13, *p* = 0.048), an association that was not observed for self-identified dyslexia. Self-identified dyslexia, however, was negatively associated with the verbal cognitive style (β = −0.23, *p* < 0.001). Self-identified dyscalculia had a significant negative association with rationality (β = −0.12, *p* = 0.02).

## 5. Discussion

Our study evaluates differences in decision-making styles, rationality, and cognitive styles among neurodivergent versus neurotypical employees by considering both formally diagnosed and self-identified conditions. We argued that employees are often aware of their neurodivergence, yet they do not seek formal diagnosis due to the inaccessibility of healthcare, high costs, or the fear of being marginalized and excluded at work ([8]). As expected, the number of self-identified neurodivergence in the sample was more than double the number of participants who reported a formal diagnosis. Given the incidence of self-identified neurodivergence, we expected (and observed) alignment between formally diagnosed and self-identified conditions in predicting the cognitive outcomes, and we also expected that the self-identified conditions would add to the predictive power of formally diagnosed conditions.

Effects of neurodivergent conditions on decision-making styles were generally aligned with previous empirical results. A formal diagnosis of ADD significantly and negatively predicted the rational decision-making style and the self-identification did not add to the predictive value of the formal one. This is consistent with prior studies that reported rational dysfunction when decision-making requires cognitive control ([19]; [25]). Moreover, consistent with the extensive decision-making strategies employed by highly gifted individuals ([2]), self-identified high giftedness was significantly associated with rational decision-making. The intuitive decision-making style was negatively predicted by education and by autism, with self-identified autism qualifying the negative association of the formal diagnosis. Such a pattern of results is due to the logical and systematic information processing characteristic of autism ([4]). Similarly, both formally diagnosed and self-identified ADHD were significantly and positively linked with spontaneous decision-making, aligning with the impulsivity associated with ADHD ([7]; [25]). The spontaneous decision-making style was also positively associated with self-identified ADD (which is explainable since ADD and ADHD share mostly similar characteristics and are often aggregated in studies) and negatively associated with formally diagnosed dyslexia. The dependent decision-making style was only predicted positively by education. The avoidant decision-making style was predicted positively by the formally diagnosed and self-identified ADD, aligning with the avoidance and procrastination failure patterns highlighted by cognitive behavioral models in prior research on this condition ([15]; [25]). The results concerning decision-making styles are aligned with the cognitive differences associated with different neurodivergent conditions and show the relevance of using self-identification in organizational studies focused on the impact of neurodivergence on work processes.

Rationality was positively predicted by education and self-identified high giftedness and negatively predicted by dyscalculia. These results are fully aligned with the cognitive performance differences reported in previous studies on the differences between dyslexia and dyscalculia ([17]) and point out the relevance of self-identified high giftedness that qualifies the effect of the formally diagnosed high giftedness (which becomes insignificant in Model 2). Concerning cognitive styles, education and ADHD positively predict the verbal style that is also negatively predicted by self-identified dyslexia. The only significant predictor of the visual style was self-identified dyslexia. The results for rationality and cognitive styles also highlight the relevance of self-identification in empirical neurodivergence research.

Future research should explore the differences between formally diagnosed and self-identified conditions, using larger data sets to allow for more robust conclusions at the level of specific conditions. A possible explanation for these differences is that scholars have noted considerable diversity in diagnostic characteristics, influenced by demographic factors ([11]). The group of formally diagnosed individuals may be more cognitively homogeneous than the self-identified group, as the need for a formal diagnosis may stem from the fact that these individuals more frequently experience deficits in cognitive tasks or social settings. In addition, larger data sets may also enable future research to explore the impact of co-occurrence of neurodivergent conditions on the predictive value of self-identification.

## 6. Limitations

The first limitation of our paper is the self-reported nature of the data. We asked participants to report both their formally diagnosed and self-identified neurodivergent conditions. Future research could use independent clinical evaluations to distinguish between formal diagnoses and self-identification in research on neurodivergence, as well as whether self-identification is actually accurate. A second limitation concerns the fact that the diagnostic conditions added in the second regression model had considerable overlap with the formal diagnosis introduced in the first step of the regression analysis. Our multicollinearity scores did not point out significant problems related to multicollinearity, yet this overall remains a boundary condition of our research design. A third limitation concerns the generalizability of our findings across different work contexts and countries. While our data encompasses nineteen organizations across seven different industries, these industries are less focused on low-skilled jobs. Since our data were collected in The Netherlands and Bonaire, this might limit the generalization to other countries, such as those with lower levels of awareness and acceptance of neurodivergent conditions. Finally, as previously noted, our study does not contain extensive demographic data, limiting our ability to control for these variables.

## 7. Conclusions

Overall, the addition of self-awareness of neurodivergent conditions added significantly to the predictive models for three decision-making styles (intuitive, avoidant, and spontaneous), both cognitive styles and rationality. Therefore, we can state that assessing neurodivergence based on self-awareness (self-identification) is a valid strategy for research purposes related to neurodivergence in organizational settings. Relying on self-identified conditions increases the incidence of neurodivergence reported in the workplace and allows for a more realistic estimation of neurodivergence in various social settings. We do not plea for the absolute superiority of self-identification over the formal diagnosis of neurodivergence, as the clinical competencies, skills, and knowledge of lay persons cannot match those of trained clinicians. However, our results show that in empirical research, self-identification can accurately capture some cognitive differences ascribed to neurodivergence.

## Figures and Tables

**Table 1 behavsci-15-00420-t001:** Means, standard deviations, and correlations.

	Mean	SD	1	2	3	4	5	6	7	8	9	10	11	12	13	14	15	16	17	18	19	20
1. Education	3.356	1.511	1																			
2. FD—High Giftedness	0.02	0.139	0.074	1																		
3. FD—Autism	0.01	0.118	0.051	−0.017	1																	
4. FD—ADHD	0.04	0.207	0.101	0.067	0.089	1																
5. FD—ADD	0.02	0.148	0.040	−0.021	0.143 **	0.150 **	1															
6. FD—Dyslexia	0.04	0.194	−0.038	−0.029	0.099	0.096	−0.031	1														
7. SI—High Giftedness	0.07	0.260	0.205 **	0.349 **	−0.033	0.148 **	0.030	−0.057	1													
8. SI—Autism	0.07	0.260	0.034	0.038	0.334 **	0.096	0.030	−0.001	0.170 **	1												
9. SI—ADHD	0.07	0.251	0.092	0.123 *	−0.032	0.428 **	0.111 *	0.061	0.226 **	0.269 **	1											
10. SI—ADD	0.08	0.274	−0.023	0.032	−0.035	−0.015	0.232 **	−0.007	0.074	0.193 **	0.330 **	1										
11. SI—DCD.	0.00	0.053	0.058	−0.007	−0.006	0.245 **	0.350 **	−0.011	0.189 **	−0.015	0.197 **	0.178 **	1									
12. SI—Dyslexia	0.08	0.274	−0.077	0.032	0.052	0.134 *	−0.045	0.574 **	−0.004	0.035	0.043	−0.013	−0.016	1								
13. SI—Dyscalculia	0.02	0.129	0.070	0.139 **	−0.016	0.077	−0.020	0.198 **	0.047	0.047	0.052	0.041	−0.007	0.121 *	1							
14. DM—Rational	0.596	3.834	−0.019	−0.011	−0.037	−0.082	−0.276 **	−0.010	0.069	0.046	−0.047	−0.102	−0.186 **	−0.042	0.018	1						
15. DM—Intuitive	0.671	3.431	−0.101	−0.007	−0.148 **	0.095	0.032	−0.065	−0.058	−0.193 **	0.098	0.099	0.045	0.029	−0.032	−0.270 **	1					
16. DM—Dependent	0.765	3.377	0.115 *	0.036	−0.040	0.081	0.034	0.014	−0.054	−0.057	0.090	0.047	0.002	0.033	0.067	0.006	−0.062	1				
17. DM—Avoidant	0.953	2.441	0.134 *	0.126 *	−0.055	0.139 **	0.192 **	0.022	0.131 *	0.133 *	0.217 **	0.295 **	0.109 *	0.048	0.109 *	−0.101	−0.042	0.410 **	1			
18. DM—Spontaneous	0.647	2.462	0.107 *	0.080	−0.004	0.239 **	0.150 **	−0.077	0.040	0.083	0.276 **	0.188 **	0.159 **	0.003	−0.012	−0.339 **	0.262 **	0.072	0.130 *	1		
19. CS—Verbal	1.770	4.087	0.180 **	0.050	0.008	0.051	−0.050	−0.092	0.133 *	0.041	0.170 **	0.026	0.087	−0.212 **	−0.031	0.087	0.038	−0.081	−0.093	−0.051	1	
20. CS—Visual	2.468	8.177	−0.008	−0.027	0.079	0.061	0.097	0.155 **	−0.020	0.076	0.031	0.083	−0.047	0.228 **	0.097	−0.099	0.084	0.052	0.190 **	0.154 **	−0.669 **	1
21. CRT	1.104	1.894	0.196 **	0.124 *	−0.010	−0.028	−0.071	0.059	0.183 **	0.027	0.036	0.103	0.005	0.029	−0.066	0.104 *	−0.138 **	−0.103	−0.026	−0.019	0.061	−0.019

Note. FD = formal diagnosis of a neurodivergent condition; SI = self-identified neurodivergent condition; DM = decision-making; CS = cognitive style; CRT = Cognitive Reflection Test; ** *p* < 0.01 and * *p* < 0.05.

**Table 2 behavsci-15-00420-t002:** Results of the regression analyses for decision-making styles.

	Rational		Intuitive		Dependent		Avoidant		Spontaneous	
	Model 1	Model 2	Model 1	Model 2	Model 1	Model 2	Model 1	Model2	Model 1	Model 2
Constant	3.87 *** (0.08)	3.89 *** (0.08)	3.59 *** (0.09)	3.57 *** (0.09)	3.18 *** (0.10)	3.15 *** (0.10)	2.14 *** (0.12)	2.04 *** (0.12)	2.32 *** (0.08)	2.27 *** (0.08)
*Formal diagnosis (step 1)*										
Education	−0.002 (0.02)	−0.01 (0.02)	−0.05 * (0.02)	−0.04 ^†^ (0.02)	0.06 * (0.03)	0.06 * (0.03)	0.07 * (0.03)	0.07 * (0.03)	0.03 (0.02)	0.04 ^†^ (0.02)
High giftedness	−0.06 (0.22)	−0.24 (0.24)	−0.05 (0.25)	−0.004 (0.27)	0.13 (0.29)	0.24 (0.32)	0.78 * (0.35)	0.55 (0.37)	0.28 (0.24)	0.34 (0.25)
Autism	0.04 (0.27)	−0.09 (0.29)	−0.87 ** (0.30)	−0.38 (0.32)	−0.37 (0.35)	−0.18 (0.38)	−0.80 ^†^ (0.42)	−0.87 (0.44)	−0.19 (0.29)	−0.12 (0.31)
ADHD	−0.11 (0.15)	−0.09 (0.17)	0.39 * (0.17)	0.32 ^†^ (0.20)	0.25 (0.20)	0.20 (0.23)	0.44 ^†^ (0.24)	0.37 (0.27)	0.69 *** (0.16)	0.49 ** (0.18)
ADD	−1.09 *** (0.21)	−0.91 *** (0.23)	0.17 (0.24)	0.02 (0.26)	0.15 (0.28)	0.14 (0.30)	1.23 *** (0.34)	0.91 * (0.35)	0.51 * (0.23)	0.27 (0.24)
Dyslexia	−0.05 (0.16)	0.08 (0.20)	−0.23 (0.18)	−0.45 * (0.22)	0.07 (0.21)	−0.12 (0.26)	0.17 (0.25)	−0.02 (0.30)	−0.29 ^†^ (0.17)	−0.42 * (0.21)
*Self-identification (step 2)*										
High giftedness		0.28 * (0.13)		−0.14 (0.15)		−0.33 ^†^ (0.18)		0.07 (0.21)		−0.23 (0.14)
Autism		0.13 (0.13)		−0.57 *** (0.15)		−0.22 (0.18)		0.32 (0.21)		0.05 (0.14)
ADHD		−0.01 (0.15)		0.28 (0.18)		0.25 (0.20)		0.14 (0.23)		0.40 * (0.16)
ADD		−0.11 (0.13)		0.27 ^†^ (0.14)		0.12 (0.17)		0.80 *** (0.19)		0.28 * (0.13)
DCD		−1.26 ^†^ (0.65)		−0.12 (0.73)		−0.44 (0.86)		−0.29 (0.99)		0.73 (0.67)
Dyslexia		−0.15 (0.14)		0.23 (0.16)		0.13 (0.18)		0.17 (0.21)		0.15 (0.15)
Dyscalculia		0.10 (0.25)		−0.09 (0.28)		0.30 (0.33)		0.47 (0.38)		−0.16 (0.26)
N										
R^2^	0.08	0.11	0.05	0.11	0.02	0.05	0.09	0.17	0.09	0.15
F change	4.95 ***	1.54	3.11 **	3.09 **	1.34	1.23	5.49 ***	4.77 ***	5.78 ***	3.14 **

Note. All neurodivergent conditions were entered as dummy variables (1 = presence, 0 = absence) with neurotypical employees as a general reference category; unstandardized regression coefficients are shown with standard errors between parentheses. *** *p* < 0.001 ** *p* < 0.01, and * *p* < 0.05. ^†^
*p* < 0.10.

**Table 3 behavsci-15-00420-t003:** Results of the regression analyses for rationality and cognitive styles.

	CRT		Visual Cognitive Style		Verbal Cognitive Style	
	Model 1	Model 2	Model 1	Model 2	Model 1	Model 2
Constant	1.40 *** (0.14)	1.36 *** (0.14)	8.10 *** (0.32)	7.85 *** (0.32)	3.44 *** (0.23)	3.59 *** (0.23)
*Formal diagnosis (step 1)*						
Education	0.15 *** (0.04)	0.14 *** (0.04)	−0.02 (0.09)	0.01 (0.09)	0.20 *** (0.06)	0.16 * (0.06)
High giftedness	0.90 * (0.41)	0.66 (0.44)	−0.37 (0.94)	−0.76 (1.00)	0.39 (0.67)	0.16 (0.70)
Autism	−0.11 (0.49)	−0.01 (0.53)	1.01 (1.12)	0.72 (1.20)	0.21 (0.80)	0.56 (0.85)
ADHD	−0.28 (0.28)	−0.21 (0.32)	0.38 (0.64)	0.49 (0.72)	0.42 (0.46)	−0.17 (0.51)
ADD	−0.48 (0.39)	−0.75 ^†^ (0.42)	1.49 ^†^ (0.89)	1.91 * (0.95)	−0.82 (0.64)	−1.32 ^†^ (0.67)
Dyslexia	0.42 (0.20)	0.59 (0.36)	1.90 ** (0.67)	0.23 (0.82)	−0.84 ^†^ (0.48)	0.39 (0.58)
*Self-identification (step 2)*						
High giftedness		0.58 * (0.25)		−0.04 (0.56)		0.42 (0.39)
Autism		−0.04 (0.25)		0.36 (0.56)		−0.08 (0.40)
ADHD		−0.16 (0.28)		−0.22 (0.63)		1.18 ** (0.45)
ADD		0.56 (0.23)		0.70 (0.53)		−0.11 (0.37)
DCD		−0.07 (1.20)		−4.74 ^†^ (2.70)		2.62 (1.91)
Dyslexia		0.004 (0.26)		1.89 ** (0.56)		−1.50 *** (0.41)
Dyscalculia		−1.03 * (0.46)		1.32 (1.03)		−0.45 (0.72)
N						
R^2^	0.06	0.11	0.04	0.09	0.05	0.12
F change	3.96 ***	2.41 *	2.32 *	2.67 *	2.86 *	4.01 ***

Note. All neurodivergent conditions were entered as dummy variables (1 = presence, 0 = absence) with neurotypical employees as a general reference category; unstandardized regression coefficients are shown with standard errors between parentheses. *** *p* < 0.001 ** *p* < 0.01, and * *p* < 0.05. ^†^ *p* < 0.10.

## Data Availability

The database for this research was submitted with the paper and is available from the corresponding author upon motivated request.

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
