# Peer review of "Formal Versus Self-Identified Neurodivergence: A Comparative Study in Work Environments"

_behavsci, 2025, doi:10.3390/bs15040420_

Round 1

Reviewer 1 Report

Comments and Suggestions for Authors

An interesting article. The need to look into self-identification is a good one. Improvements:

  • Justification of why education was used as a control variable
  • Avoid claims that the study compares neurodivergence to neurotypicalness- there's no data on neurotypicals in the study- so what does this mean for the findings?
  • Expand the limitations to include the generalizability/ application of findings across work contexts
  • Strengthen the arguments/discussion around strengths of different cognitive styles and the different conditions
  • Address the impact of co-occurrence on the findings and the way you coded the data

Author Response

Comments and Suggestions for Authors

An interesting article. The need to look into self-identification is a good one.

Answer: Thank you very much for reviewing the manuscript and seeing the value of this study. We highly appreciate your willingness to provide feedback and believe that your feedback has increased the quality of the manuscript.

Improvements:

Justification of why education was used as a control variable

Answer: Thank you for highlighting the importance of further justifying the inclusion of educational level as a control variable. In response to this feedback, we made two adjustments to the manuscript. First, we expanded the methods section to clarify how educational level was measured, which reads as follows:

Education was evaluated as a categorical variable by asking participants to specify their highest completed level of education: (1) vocational education; (2) secondary education; (3) bachelor's degree from a university of applied sciences; (4) master's degree from a university of applied sciences or bachelor's degree from a research university; (5) master's degree from a research university; (6) and doctoral degree.

Additionally, we briefly explained the relevance of educational level in the context of cognitive functioning, supported by a reference to the study of Guerra-Carrillo et al. (2017):

“To preserve anonymity, we did not collect demographic information, except for education that was deemed relevant as a control variable because educational achievements can be related to the neurodivergent conditions and because education is correlated with cognitive competencies, cognitive functioning and information processing styles (Guerra-Carrillo et al., 2017).”

.

Avoid claims that the study compares neurodivergence to neurotypicalness- there's no data on neurotypicals in the study- so what does this mean for the findings?

Answer: Thank you for pointing this out. We followed your suggestion and toned down the comparison claim. However, the sample did include neurotypical individuals as presented in the sample description. The revised Methods section clarifies this issue and also emphasizes that the results are aligned with previous research on self-diagnosis of neurodivergence (Fellowes, 2024). In the dummy coding of our variables the neurotypical category is implicitly entered into the analyses as the baseline comparison category with all the neurodivergent conditions. However, the key contribution of our study is the predictive power added by the self-identified neurodivergent conditions to the formal diagnosed ones. We keep this as the key contribution of our paper.  

Expand the limitations to include the generalizability/ application of findings across work contexts

Answer: We followed your suggestion and expanded the limitations section to address the generalizability across work contexts. In addition, following another reviewer’s feedback, we also added a limitation regarding the generalizability across countries, as our data was collected solely in the Netherlands and Bonaire. The added text reads as follows:

A third limitation concerns the generalizability of our findings across different work contexts and countries. While our data encompasses nineteen organizations across seven different industries, these industries are less focused on low-skilled jobs. And since our data was collected in The Netherlands and Bonaire, this might limit the generalization to other countries, such as those with lower levels of awareness and acceptance of neurodivergent conditions.

Strengthen the arguments/discussion around strengths of different cognitive styles and the different conditions

Answer: Thank you for pointing this out. We discuss the key findings in various sections of the discussion and we emphasize the strengths and weaknesses tied to the neurodivergent conditions putting it in perspective with previous research. However, we do not want to be too speculative and stretch the interpretation of our results beyond what the data shows and beyond what is reasonable for the sample we used. We hope you concur with us on this point.

Address the impact of co-occurrence on the findings and the way you coded the data

Answer: We agree that the co-occurrence of multiple neurodivergent conditions may affect our results in a certain way. However, our dataset is too small to test these implications. Therefore, in response to your feedback, we decided to bring this point to the table as a suggestion for future research at the end of our discussion chapter, where we have already discussed the possibilities for larger datasets. Our addition reads as follows:

In addition, larger data sets may also enable future research to explore the impact of co-occurrence of neurodivergent conditions on the predictive value of self-identification.

References

Guerra-Carrillo, B., Katovich, K., & Bunge, S. A. (2017). Does higher education hone cognitive functioning and learning efficacy? Findings from a large and diverse sample. PloS One, 12(8). https://doi.org/10.1371/journal.pone.0182276

Reviewer 2 Report

Comments and Suggestions for Authors

This is a novel and interesting paper, and its consideration of forms of neurodivergence beyond just autism and ADHD is a strength.  

It's unfortunate that detailed demographic data were not available, as it would be interesting to look at rates of self-diagnosis between younger/older employees.  However, the authors do note this as a limitation. 

The only demographic detail collected was education. However, it is not specified how education was measured (years of education, degree vs no degree?) and thus how it was included in the model.  The authors also state it was necessary to control for education in relation to "cognitive competencies and information processing styles", but do not seem to cite any paper establishing a link between these.

The authors note that the sample reflects "357 participants across 19 organizations in 7 industries".  To help contextualize the study setting, the authors should provide details about these 7 industries (are they quite diverse) and whether these industries were specifically selected. 

Furthermore, as all participants lived/worked in the Netherlands or Bonaire, the authors should address whether they expect these findings to generalize to other countries/regions. 

Lastly, some minor typos were noted while reading the paper. The overall writing is clear, but final proofreading is needed.

Author Response

Reviewer 2

Comments and Suggestions for Authors

This is a novel and interesting paper, and its consideration of forms of neurodivergence beyond just autism and ADHD is a strength.  

It's unfortunate that detailed demographic data were not available, as it would be interesting to look at rates of self-diagnosis between younger/older employees.  However, the authors do note this as a limitation. 

Answer: Thank you very much for your willingness to read and review the manuscript. We are glad to read that you support the added value of the study. We share your view that it is unfortunate that further demographic data is absent. But as you point out, this is included in the paper as a limitation, which stems from ethical considerations.

The only demographic detail collected was education. However, it is not specified how education was measured (years of education, degree vs no degree?) and thus how it was included in the model. The authors also state it was necessary to control for education in relation to "cognitive competencies and information processing styles", but do not seem to cite any paper establishing a link between these.

Answer: We agree with you that this part was not yet sufficiently described. Based on your feedback, we have made two adjustments. First, we have included in the method section educational level as a variable with a description of the categories, which reads as follows:

Education was evaluated as a categorical variable by asking participants to specify their highest completed level of education: (1) vocational education; (2) secondary education; (3) bachelor's degree from a university of applied sciences; (4) master's degree from a university of applied sciences or bachelor's degree from a research university; (5) master's degree from a research university; (6) and doctoral degree.

Second, we briefly explained why education is an important control variable in the context of this study. In line with your suggestion, we also added a citation to support this consideration:

“To preserve anonymity, we did not collect demographic information, except for education that was deemed relevant as a control variable because educational achievements can be related to the neurodivergent conditions and because education is correlated with cognitive competencies, cognitive functioning and information processing styles (Guerra-Carrillo et al., 2017).”

.

The authors note that the sample reflects "357 participants across 19 organizations in 7 industries".  To help contextualize the study setting, the authors should provide details about these 7 industries (are they quite diverse) and whether these industries were specifically selected. 

Answer: We followed your suggestion and added both the names of the seven industries as well as the number of organizations per industry. No industry selection was made; instead, as stated, a snowball sampling approach was used, and the resulting diversity of industries is a natural outcome of this method.

Furthermore, as all participants lived/worked in the Netherlands or Bonaire, the authors should address whether they expect these findings to generalize to other countries/regions. 

Answer: Thank you for pointing out this limitation. We followed your suggestion and added a new limitation to the paper to address the limitation of generalization to other countries. In addition, following the suggestion of the other reviewer, we also added the possible limited generalization to other industries. Our added limitation reads as follows:

A third limitation concerns the generalizability of our findings across different work contexts and countries. While our data encompasses nineteen organizations across seven different industries, these industries are less focused on low-skilled jobs. And since our data was collected in The Netherlands and Bonaire, this might limit the generalization to other countries, such as those with lower levels of awareness and acceptance of neurodivergent conditions.

Lastly, some minor typos were noted while reading the paper. The overall writing is clear, but final proofreading is needed.

Answer: We have re-read the entire manuscript and removed some minor typos.

References

Guerra-Carrillo, B., Katovich, K., & Bunge, S. A. (2017). Does higher education hone cognitive functioning and learning efficacy? Findings from a large and diverse sample. PloS One, 12(8). https://doi.org/10.1371/journal.pone.0182276

Round 2

Reviewer 1 Report

Comments and Suggestions for Authors

The revised version is fine.